# Gastric residual volume measurement in British neonatal intensive care units: a survey of practice

Jon Dorling [ID],[1] Lyvonne Tume,[2] Barbara Arch,[3] Kerry Woolfall,[4] Lynne Latten,[5] Louise Roper [ID],[4] Elizabeth Deja,[4] Nazima Pathan,[6] Helen Eccleson,[3] Helen Hickey,[3] Michaela Brown,[3] Anne Beissel [ID],[7] Izabela Andrzejewska,[8] Frederic Valla,[9] Chris Gale [ID] [10]

► Additional material is published online only. To view please visit the journal online (http://dx.doi.org/10.1136/bmjpo-2019-000601).

**Correspondence to**
Dr Jon Dorling; Jon.Dorling@iwk.nshealth.ca

## ABSTRACT

**Objective** Despite little evidence, the practice of routine gastric residual volume (GRV) measurement to guide enteral feeding in neonatal units is widespread. Due to increased interest in this practice, and to examine trial feasibility, we aimed to determine enteral feeding and GRV measurement practices in British neonatal units.

**Design and setting** An online survey was distributed via email to all neonatal units and networks in England, Scotland and Wales. A clinical nurse, senior doctor and dietitian were invited to collaboratively complete the survey and submit a copy of relevant guidelines.

**Results** 95/184 (51.6%) approached units completed the survey, 81/95 (85.3%) reported having feeding guidelines and 28 guidelines were submitted for review. The majority of units used intermittent (90/95) gastric feeds as their primary feeding method. 42/95 units reported specific guidance for measuring and interpreting GRV. 20/90 units measured GRV before every feed, 39/90 at regular time intervals (most commonly four to six hourly 35/39) and 26/90 when felt to be clinically indicated. Most units reported uncertainty on the utility of aspirate volume for guiding feeding decisions; 13/90 reported that aspirate volume affected decisions 'very much'. In contrast, aspirate colour was reported to affect decisions 'very much' by 37/90 of responding units. Almost half, 44/90, routinely returned aspirates to the stomach.

**Conclusions** Routine GRV measurement is part of standard practice in British neonatal units, although there was inconsistency in how frequently to measure or how to interpret the aspirate. Volume was considered less important than colour of the aspirate.

## INTRODUCTION

The gastric residual volume (GRV) is the volume of the entire stomach contents, obtained by aspiration with a syringe in order to assess feeding tolerance. It provides information on the volume and colour of fluid, and is distinct from the aspiration of a small amount of fluid for pH testing to confirm feeding tube position.[1] There is a paucity of evidence to support routine measurement of GRV to direct and guide enteral feeding, and

### What is known about the subject?

► The gastric residual volume (GRV) is the volume of the entire stomach contents, obtained by aspiration with a syringe in order to assess feeding tolerance.

► It is unclear if the routine measurement of GRV is beneficial or harmful in preterm infants.

► Those who routinely measure GRV are attempting to identify necrotising enterocolitis early and aim to prevent complications by withholding or reducing feed volumes.

### What this study adds?

► This study demonstrates mixed practice for residual measurements across neonatal units in Britain.

► Aspirate colour was reported as affecting decisions more often in comparison to residual volume.

► A randomised trial appears feasible in Great Britain given the variation in practice and willingness of respondents to randomise to measuring or not measuring.

the practice is increasingly being questioned in neonatal units.[1–5] For many clinicians, however, this parameter is a fundamental part of the definition and diagnosis of feed intolerance.[6] The rationale for routinely measuring GRV in the neonatal setting is for the early identification of necrotising enterocolitis (NEC) and prevention of complications such as vomiting or aspiration, by withholding or reducing feed volumes.[1 7 8] Routine measurement could, however, cause harm, for example, through direct injury of the gastric mucosa, discarding gastric juices, medications and hormones, and by delaying enteral feeding and prolonging parenteral nutrition.[4 9 10] Furthermore, measurement of GRV has been shown to be inaccurate and affected by the position of the baby and the

tube, hence it is not a useful surrogate marker for delayed gastric emptying in premature infants.[11–14]

In this study, we aimed to identify current practice around GRV measurement in Great Britain. In addition, we sought to delineate enteral feeding practices in UK neonatal units in relation to GRV, and to identify a 'control arm' for a future trial comparing no routine GRV measurement (the intervention) to routine GRV measurement using this and work published elsewhere.

## METHODS

A survey instrument was developed by the research team to explore current practices around GRV measurement and general enteral feeding practices in neonatal units. The intention was to use these survey findings alongside a review of neonatal unit guidelines to establish current practice. A 10-item closed question survey (tick-box responses) with optional free text response and nine open-ended questions was developed by the researchers. The survey was piloted for face validity with 10 staff (doctors, dietitians, nurses). Minor wording adjustments were made to improve clarity, before the 19-item survey (online supplementary material) was entered onto the survey platform and retested by the study team.

The survey focused on three domains: general enteral feeding and nutrition practices in the respondents' unit, the GRV measurement technique used in the respondents' unit and clinical management in response to GRV. The survey invitation requested that a senior doctor, a clinical nurse and a dietitian complete the survey collaboratively and submit one response per unit, and requested that any relevant written guidelines or protocols be submitted. Unit name was collected, to target non-responders and check for duplicates; three reminders were sent to maximise response rates. Our target response rate was 70%.

All National Health Service neonatal units in England, Scotland and Wales were approached during May and June 2018 using email invitations directed at 184 neonatal teams (some neonatal teams cover multiple neonatal units) sent through a national research collaboration, UK Neonatal Collaborative (UKNC), and a multidisciplinary professional network, the Neonatal Nutrition Network. Units in Northern Ireland were not contacted as they are not part of the UKNC. Study data were collected and managed using REDCap electronic data capture tools hosted at the University of Liverpool.[15] Data were summarised using descriptive statistics for quantitative data and a mix of thematic and content analysis for qualitative free text data.[16 17] Following this, the neonatal unit guidelines were reviewed and summarised.

### Patient and public involvement

There was no direct patient or public involvement in the work presented in this manuscript as it involved surveying clinicians on their clinical practice. Other aspects of the research not reported here had substantial input as they involved qualitative interviews and consensus gathering.[18]

## RESULTS

Ninety-five of 184 (51.6%) neonatal units in the UK excluding Northern Ireland completed the survey (tables 1 and 2). These consisted of 40 neonatal intensive care units (NICU), 42 local neonatal units (LNU) and 13 special care baby units (SCBU) giving response rates of 71.4%, 47.2% and 33.3% of the NICUs, LNUs and SCBUs, respectively. Seventeen of a possible 18 NICUs caring for both surgical and medical patients responded, as did 23 NICUs caring for medical cases only. LNUs and SCBUs do not provide early postoperative care in the UK.

Survey responses were received from senior doctors (81/95, 85.3%), nurses (51/95, 53.7%) and dietitians (9/95, 9.5%). Most (81/95, 85.3%) responding units reported written enteral feeding guidance and 28 unit or local neonatal network guidelines were sent to the author (online supplementary table 2). Enteral feeding was typically delivered intermittently (90/95, 94.7%) rather than continuously (5/95, 5.3%). Forty-two of 95 units (44.2%) reported having written guidance for measurement and interpretation of GRVs. Ninety units answered questions about the management of non-surgical babies. When asked about how often GRV is measured, 20/90 units (22.2%) measured aspirates before every feed, 26/90 (28.9%) when it was felt to be clinically indicated and 39/90 (43.3%) measured GRV at regular time intervals (most commonly four to six hourly 35/39 (89.7%), but all more frequent than once per day). One unit had no guidelines on this, and 4/90 (4.4%) reported that they did not measure GRV. Ninety open text responses were received to the question 'Are Gastric Residual Volume measured for all babies, or just below a set gestational age/birth weight or for a specific condition?' Over 30 responses said all babies should have gastric aspirates measured, with some additional responses limiting this to those on gastric tube feeds or until full feeds are established. Just six responses mentioned a gestational age cut-off, four suggesting <32 weeks, one <27 weeks and one <34 weeks' gestation. Just one response indicated a birth weight criterion (under 1500 g at birth). Online supplementary table 1 presents the responses.

Among units that reported having written GRV measurement guidance, 13/39 (33.3%) indicated that the guidance was 'always', and 17/39 (38.6%) 'usually' followed; however, free text responses suggested that practice was '*very variable depending on the nurse looking after the baby*' (Unit 3, surgical and medical unit). The bedside nurse most commonly made decisions in relation to GRV results, 56/90 (62.2%), followed by middle grade doctors, 41/90 (45.6%), and the senior nurse in charge of shift, 26/90 (28.9%).

Responding units had mixed views on how useful the volume of the aspirate was for guiding feeding decisions (figure 1): just 13/90 (14.4%) of units reporting that

**Table 1** Survey results—GRV practices specific to the management of medical babies (n=90)

| Survey question | n (%) |
|---|---|
| **How often do staff in your unit measure GRV?** | |
| Once a day | 0 (0) |
| Before every feed | 20 (22.2) |
| Only when clinically indicated | 26 (28.9) |
| At regular intervals | 39 (43.4) |
| *At least every 3, 4 or 6 hours* | *35/39* |
| GRV is not measured | 4 (4.4) |
| **Is the specific guidance for GRV measurement followed and actually undertaken as per protocol—only asked of units with specific guidance for GRV measurement (n=39)?** | |
| Always | 13 (43.3) |
| Usually | 17 (38.6) |
| Often | 4 (10.3) |
| Rarely/Never | 5 (12.8) |
| **Who usually decides what to do with concerning GRV aspirates in the first instance? (more than one response allowed)** | |
| Senior doctor (consultant) | 13 (14.4) |
| Middle grade doctor (SpR) | 41 (45.6) |
| Junior grade doctor (SHO) | 18 (20.0) |
| Bedside nurse | 56 (62.2) |
| Nurse in charge of shift (senior nurse) | 26 (28.9) |
| **How much does volume of the aspirate affect your decision around GRV?** | |
| 1 (Not at all) | 5 (5.6) |
| 2 | 11 (12.2) |
| 3 | 40 (44.4) |
| 4 | 21 (23.3) |
| 5 (Very much) | 13 (14.4) |
| **How much does colour of the aspirate affect your decision around GRV?** | |
| 1 (Not at all) | 3 (3.3) |
| 2 | 6 (6.7) |
| 3 | 16 (17.8) |
| 4 | 28 (31.1) |
| 5 (Very much) | 37 (41.1) |
| **What do you do with obtained GRV: return or discard?** | |
| Return | 44 (48.9) |
| Discard | 7 (7.8) |
| Other | 39 (43.3) |

GRV, gastric residual volume; SHO, senior house officer; SpR, specialist registrar.

volume affected clinical decision-making 'very much' and the most frequent response was an intermediate score. The colour of the aspirate was felt to be more important: 37/90 (41.1%) of units reporting that colour influenced clinical decisions 'very much' and this was the most frequent response. More detail was obtained from 74 open text responses to this question. A large volume of aspirate was commonly described as a concern, which would often lead to a clinical review of a baby's condition and subsequent consideration of the how much milk the baby is receiving. The threshold for prompting a feeding review was reported to vary. Some units stated that aspirates over 50% of the feed would 'prompt a review' (Unit 8, NICU surgical and medical), while others stated '>25% of feed given in previous 6 hours' (Unit 18, NICU medical only), if exceeds '25% of the previous 4 hours' feed volume' (Unit 22, NICU medical only) or 'If >25% of the feed volume given since the last assessment was made' (Unit 25, NICU medical only).

Almost half, 44/90 (48.9%), routinely returned aspirates to the stomach. Seventy-two nurses gave reasons for seeking medical advice: 55/72 (76.4%) cited increased or large volume GRVs, 52/72 (72.2%) cited bilious colour of the residual, or a change in colour. Other reasons were blood-stained aspirates 16/72 (22.2%), concerns about condition of the baby, such as desaturations 16/72 (22.2%), abdominal distention 11/72 (15.3%) and vomiting 5/72 (6.9%). In free text responses, units stated that a dark or bilious colour would '*trigger medical review [by a] Middle Grade or Consultant*' (Unit 22, NICU medical only), while some described how feeds would be stopped: '*Green aspirate—assess baby and feeds withheld*' (Unit 60, LNU).

Guideline analysis (online supplementary table 2) revealed that 19 of 28 (67.8%) guidelines specified a volume of aspirate at which to consider stopping feeds using a defined proportion of the previous feed. Six guidelines specified this threshold as 25% or more of the previous feed, eight guidelines specified 50% or more, while five guidelines used a level between these. Fourteen guidelines mentioned the bilious green colouring of GRV being an indication to stop enteral feeds, while five mentioned blood staining as being important. Vomiting and abdominal distension were also considered important for guiding management being mentioned by 13 and 12 guidelines, respectively.

## DISCUSSION

The results of this survey confirm mixed practice in neonatal units across the UK for both monitoring GRV and in how findings are used to make decisions about enteral feeding. This survey also identifies that around half of British neonatal units use GRV as a parameter to guide enteral feeding advancement. Health professionals' views around the importance of the volume compared with the colour of the GRV were inconsistent and importance was defined at different thresholds. Aspirate colour was cited more often as important than volume of gastric residuals; however, the importance of aspirate colour was inconsistent; some unit guidelines

**Table 2** Survey results—general feeding practices for all babies (n=95)

| Practice | n (%) |
| --- | --- |
| Units had written feeding guidelines/protocol. | 81 (85.3) |
| Standard NG feeds were intermittent bolus (not continuous). | 90 (94.7) |
| There was specific guidance about how gastric residual volume should be measured and interpreted—for example, a protocol or guideline. | 42 (44.2) |
| NICUs that care for surgical and medical babies (n=17) | |
| Gastric residual volume measurement differs between the medical and surgical babies. | 5/17 (29.4) |

NG, nasogastric; NICU, neonatal intensive care unit.

specified actions based on bilious or blood staining of the secretions whereas others did not mention them, and many unit guidelines referred to not returning aspirates that were bilious (green) or bloody (red) in colour. Change in aspirate colour was viewed as a potential indicator of NEC in preterm neonates in this survey, but this and many aspects of residual evaluation are unsubstantiated by high-quality evidence.[5]

The mixed views elicited on interpreting volume are consistent with the paucity of evidence for routine GRV measurement, and support randomised trials to assess whether aspirating the stomach contents is a useful practice.[4 7 9] Such a trial is also supported by a recently published Cochrane review.[1] Although it might be beneficial to stop measuring GRV in neonatal units, some health professionals believe their measurement can help identify NEC earlier despite the absence of evidence to support this presumption. Recent results from small studies involving preterm infants suggest that not measuring GRV is not associated with an increase in the risk of NEC and might reduce the time to achieve full enteral feeds[3 4 9 19]; however, these studies were underpowered to detect even large relative differences in rare outcomes like NEC. Adequate power to definitively assess NEC would require a trial of thousands of participants rather than the 230 randomised participants studied to date.[2 3] Routine monitoring of GRV does however add to nursing workload and may lead to other direct harms to the infant. Given the widespread use of this practice, a future trial would need to demonstrate the safety of both routinely monitoring and not monitoring GRV. Further

details of the proposed trial are published elsewhere (NIHR HTA journal in press).

The routine measurement of GRV is based on the presumption that GRVs are an accurate representation of the residual gastric contents. Laboratory-based simulation studies undermine this presumption, however, by demonstrating that GRV inaccurately measures gastric contents.[20 21] The GRV obtained is widely influenced by a number of factors such as the syringe size, gastric tube size and material, aspiration pressure, viscosity of aspirate and both the position of the tube tip in the stomach and of the neonate.[22] Furthermore, when decision-making is based on volume, clinicians fail to consider the impact of gastric secretions produced during the digestion process.[23]

This study has limitations. First, as with any survey, responses may not reflect actual practice. However, we were able to obtain a summary of what ought to happen by reviewing unit guidelines. Second, it is a weakness of the study that there were low responses from the smaller neonatal units. The results might therefore overrepresent the views of larger NICU units. Third, we asked an open rather than a closed question to seek detail on which babies (in terms of gestation, birth weight or conditions) have residual volumes measured which made the data hard to analyse. Further details were obtained in related research and have been published elsewhere (NIHR HTA journal in press).

## CONCLUSIONS

The routine and frequent measurement of GRV is embedded in enteral feeding practice and guidelines in British neonatal units, despite a lack of evidence and questionable accuracy of this parameter. For many units, GRV is integral to the assessment of feed tolerance/intolerance with bilious colouring of the aspirate and presence of blood being considered important. This study has identified current practice around GRV measurement in British neonatal units, and supports examination of the benefits and harms of GRV in an adequately powered randomised controlled trial.

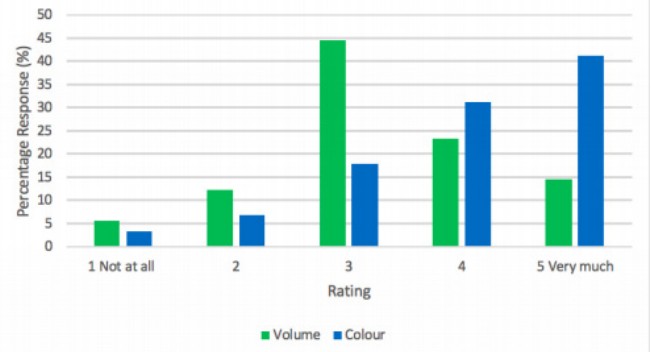

**Figure 1** Perceived importance of aspirate volume and colour for making feeding decisions.

**Author affiliations**
[1]Division of Neonatal-Perinatal Medicine, Dalhousie University Faculty of Medicine, Halifax, Nova Scotia, Canada

 Dorling J, *et al. BMJ Paediatrics Open* 2020;**4**:e000601. doi:10.1136/bmjpo-2019-000601

[2]Child Health, University of Salford, Salford, UK
[3]Liverpool Clinical Trials Unit, University of Liverpool, Liverpool, UK
[4]Health Services Research, University of Liverpool, Liverpool, UK
[5]Dietetics, Alder Hey Children's Hospital, Liverpool, UK
[6]Paediatrics, University of Cambridge, Cambridge, UK
[7]Neonatal Intensive Care Unit, Hôpital Femme Mère Enfant, Lyon-Bron, France
[8]Neonatal Unit, Chelsea and Westminster Healthcare NHS Trust, London, UK
[9]Service de Réanimation Pédiatrique, Hôpital Femme-Mère-Enfant, Hospices Civils de Lyon, Université Claude-Bernard Lyon 1, Lyon, France
[10]Academic Neonatal Medicine, Imperial College London, London, UK

**Acknowledgements** We thank all the neonatal units who took part in this survey. This publication contains information also published in the NIHR HTA journal under an agreement that this acknowledgement is made.

**Contributors** JD and LT planned the study and incorporated comments from all members on the study design, conduct, interpretation and reporting of the work. BA analysed the quantitative data. KW, ED and LR analysed the open answer responses. LT and JD analysed the guideline content. JD is responsible for the overall content as guarantor.

**Funding** This work was supported by NIHR HTA programme as part of a larger NIHR HTA-funded feasibility study (grant number 16/94/02).

**Disclaimer** The views expressed are those of the authors and not necessarily those of the NHS, the NIHR or the Department of Health and Social Care.

**Competing interests** JD reports grants from National Institute for Health Research (NIHR), during the conduct of the study; grants from NIHR, grants from Nutrinia, outside the submitted work. LT reports grant from NIHR, during the conduct of the study. ED reports grants from NIHR Health Technology Assessment (HTA) programme during the conduct of the study. HE reports grants from NIHR HTA during the conduct of the study. HH reports this grant from NIHR HTA, during the conduct of the study. MB reports grants from NIHR HTA during the conduct of the study. CG reports grants from NIHR, during the conduct of the study; grants from NIHR, grants from Medical Research Foundation, grants from Mason Medical Research Foundation, grants and personal fees from Chiesi Pharmaceuticals, grants from Rosetrees Foundation, grants from Canadian Institute for Health Research, outside the submitted work. FV reports personal fees from Baxter, personal fees from Nutricia, outside the submitted work. LT reports grants from NIHR during the conduct of the study.

**Patient consent for publication** Not required.

**Ethics approval** Ethical approval for the study was provided by the University of the West of England (Reference: HAS.18.04.144).

**Provenance and peer review** Not commissioned; externally peer reviewed.

**Data availability statement** All data relevant to the study are included in the article or uploaded as supplementary information.

**ORCID iDs**
Jon Dorling http://orcid.org/0000-0002-1691-3221
Louise Roper http://orcid.org/0000-0002-2918-7628
Anne Beissel http://orcid.org/0000-0002-1033-451X
Chris Gale http://orcid.org/0000-0003-0707-876X

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
