## [Reviewer comments · BMJ Paediatrics Open]

ARTICLE DETAILS

TITLE (PROVISIONAL)	Gastric Residual Volume measurement in British neonatal intensive care units: a survey of practice
AUTHORS	Dorling, Jon; Tume, Lyvonne; Arch, Barbara; Woolfall, Kerry; Latten, Lynne; Roper, Louise; Deja, Elizabeth; Pathan, Nazima; Eccleson, Helen; Hickey, Helen; Brown, Michaela; Beissel, Anne; Andrzejewska, izabela; Valla, Frederic; Gale, Chris

VERSION 1 – REVIEW

REVIEWER	Reviewer name: Dr Thangaraj Abiramalatha Institution and Country: Sri Ramachandra Institute of Higher Education and Research, Chennai, India Competing interests: None
REVIEW RETURNED	03-Mar-2020

GENERAL COMMENTS	I congratulate the authors for choosing a relevant topic and doing a reasonably good descriptive survey on the practice of gastric residual monitoring in the neonatal units in UK. Though there is no concrete evidence that gastric residual monitoring predicts and prevents NEC, it is widely practiced in many centres across the world. It is good to know the prevalence of such practice, which would help us to identify knowledge gaps and plan future clinical trials. There is not much existing literature on the prevalence of the practice of gastric residual monitoring, which makes this study interesting. The article needs some revisions before it can be considered for publication. Major comments Methods 1. Clinical trial registration details need to be given.2. 19-item survey (supplementary material) is missing3. Infant demographics need to be described. GR monitoring in which group of infants? Gestational age/ birth weight?4. How was the validity and reliability of the questionnaire tested? From the description, I understand only pre-testing was done. Any pilot testing or clinical sensibility testing done? The authors are referred to the article “A guide for the design and conduct of self-administered surveys for clinicians” by Burns et al (DOI:10.1503/cmaj.080372) - for the steps of developing a survey questionnaire. And identify the missing steps as limitations.5. Did the authors calculate sample size? Though the authors intended to include a minimum of 70% the 184 centers in NHS, only 95 (51.6%) units responded. Whenever the response rate is <70%, external validity (generalizability) of the results is questionable.
---

	Did the authors take additional efforts to improve the response rate, other than the 3 reminders? – such as precontact, personalized emails, request for explanation for non-participation etc. Discussion 1. Other surveys addressing gastric residuals (for instance, the article with PMID: 22301544) should be compared and discussed. Minor comments Abstract 1. ‘Gastric residual volume (GRV) measurement’ can be changed to ‘gastric residual (GR) monitoring’. Because, not only the volume but the quality of GR is also monitored. Background 1. ‘Gastric residual volume (GRV) measurement’ can be changed to ‘gastric residual (GR) monitoring’ 2. The rationale of the study may better be written like this: “There are a few RCTs and a recent Cochrane meta-analysis (PMID 31425604) on no monitoring versus routine monitoring of gastric residual (references), which suggest that the available evidence is inadequate. Hence, we are planning to do a multicentric RCT on the same. This survey was conducted to...” Results 1. Results can be written much more concise. There is no need to repeat all the data given in table 2. Discussion 1. The second limitation can be rephrased as “low response rate.... affects the external validity (generalizability) of the study results” What this study adds 1. Points 2 & 3 need to be rephrased
--	--

REVIEWER	Reviewer name: Ariel Salas Institution and Country: University of Alabama at Birmingham Competing interests: None related to this manuscript
REVIEW RETURNED	09-Mar-2020

GENERAL COMMENTS	With this survey, the authors attempted to summarize current feeding practices concerning gastric residual volumes in preterm infants. The authors conducted this research to examine the feasibility of a future trial with a study arm in which clinicians do not measure gastric residual volumes. The conclusion statement suggests that a randomized trial with such characteristics is feasible. Although this reviewer may not agree on all the issues raised in the article, I praise the authors for their efforts on describing current feeding practices in the UK. Below I summarized my specific comments and suggestions.  - This article would have been more persuasive if the authors had related their findings with NEC outcomes of the units included in this survey. Specifically, it would have been interesting to know if units that do not currently measure GRVs have a lower incidence of NEC or culture-proven sepsis. - The abstract summarizes well the results of the survey. - The introduction section is balanced and well-written. - A correlation analysis between survey responses of senior doctors
--

	and survey responses of nurses from the same unit would help validate the accuracy of their responses. In survey-based methodology, some suspect that senior doctors report what they think they do and nurses and other front-line clinicians report what they really do.  - In the discussion section, it would be helpful to make the distinction between practices that promote routine GRV measurements and practices that promote selective GRV measurements. The practice of not performing GRV measurements at all should be discussed separately. - One aim of the survey was to define the control arm of a future trial in which GRVs are not measured. The characteristics of this potential control arm were not reported. Based on the study findings and the limited evidence available, it would seem reasonable to have a control arm with selective GRV measurements. In this group, GRVs would be routinely returned regardless of color to maintain a low gastric pH and minimize the risk of bacterial colonization. - The proposed trial implies an effort to standardize the technique of gastric content aspiration among preterm infants. A clear efficacy endpoint of checking GRVs to prevent adverse outcomes of preterm infants should be stated to make the safety endpoint relevant. As currently written, the justification for a trial with safety as the primary outcome is weak.
--	---

VERSION 1 – AUTHOR RESPONSE

Reviewer 1:

“I congratulate the authors for choosing a relevant topic and doing a reasonably good descriptive survey on the practice of gastric residual monitoring in the neonatal units in UK. Though there is no concrete evidence that gastric residual monitoring predicts and prevents NEC, it is widely practiced in many centres across the world. It is good to know the prevalence of such practice, which would help us to identify knowledge gaps and plan future clinical trials. There is not much existing literature on the prevalence of the practice of gastric residual monitoring, which makes this study interesting. The article needs some revisions before it can be considered for publication.”

Reviewer 1 Comments		Response
Major		
1	Clinical trial registration details need to be given.	The study was not a clinical trial, it is a survey of practice. It is part of a larger feasibility study whose purpose is to design a future clinical trial. Please see http://grvstudy.com/
2	19-item survey (supplementary material) is missing	The survey was previously uploaded and has been uploaded again as a supplemental file titled 'NICU Gastric Survey Questions'.
3	Infant demographics need to be described. GR monitoring in which group of infants? Gestational age/ birth weight?	The study is a survey of practice, and the population of interest was all neonatal intensive care units in the UK. The population of neonates cared for within these units was not described, as our primary question was; 'what is the current practice regarding the measurement of gastric residual volume in NNUs across the UK'?
4	How was the validity and reliability of the questionnaire tested? From the description, I	The questionnaire was designed to be completed by a team of clinicians from

	understand only pre-testing was done. Any pilot testing or clinical sensibility testing done? The authors are referred to the article “A guide for the design and conduct of self-administered surveys for clinicians” by Burns et al (DOI:10.1503/cmaj.080372) - for the steps of developing a survey questionnaire. And identify the missing steps as limitations.	each unit, including a doctor, a nurse, and a dietician. In this way, we obtained a broad reflection of current practice from each participating unit. We also collected units' guidelines to compare whether what they said they did was similar to what their guidelines stated. In our Methods section we describe that the survey was piloted to ensure it made sense. As we were not attempting to measure a subjective continuous outcome as a result of the responses, and the survey is not being proposed as a measurement tool for future studies, we do not consider that checks for validity and reliability are required.
	Did the authors calculate sample size? Though the authors intended to include a minimum of 70% the 184 centers in NHS, only 95 (51.6%) units responded. Whenever the response rate is <70%, external validity (generalizability) of the results is questionable.	A sample size calculation was not carried out. The purpose of the study was not to draw inference, but to obtain descriptive statistics. We consider the response rate to have been excellent, given that the survey was voluntary, and that a team of clinicians was needed in order to complete the survey. The results would be limited if it were not representative of UK units in general – however, we have shown that the sample is reasonably representative of the three types of neonatal unit targeted. We include comments on this in our discussion.
6	Discussion: Other surveys addressing gastric residuals (for instance, the article with PMID: 22301544) should be compared and discussed.	The referenced article describes North American feeding practice more generally and is located behind a paywall. The abstract does not mention GRV. Our paper is intended to be relevant to the UK and describes practice there and the potential need for a trial in the UK. It could be interesting to describe other countries too but would take away from the focus of our study.
Minor		
1	Abstract: ‘Gastric residual volume (GRV) measurement’ can be changed to ‘gastric residual (GR) monitoring’. Because, not only the volume but the quality of GR is also monitored.	Thank you – the term GRV was used in the wider GASTRIC feasibility study, and was the terminology used in our e-survey. We accept when GR is measured, the colour is also looked at, but would like to keep the wording unchanged.
2	Background: ‘Gastric residual volume (GRV) measurement’ can be changed to ‘gastric residual (GR) monitoring’	See previous response.
3	Background: The rationale of the study may better be written like this: “There are a few RCTs and a recent Cochrane meta-analysis (PMID 31425604) on no monitoring versus routine monitoring of gastric residual (references), which suggest that the available evidence is inadequate. Hence, we are planning to do a multicentric RCT on the same. This survey was conducted to...”	Thank you. We feel that the rationale for the study has been explained. The word ‘rationale’ may have given confusion though – we have used it to talk about the rationale for GRV in current practice, rather than regarding the rationale for the study. We thank the reviewer for highlighting the recent Cochrane review publication which

		has been included as reference 1 and referred to in the discussion as “Such a trial is also supported by a recently published Cochrane review(1)”. The introduction has also been amended to make it clearer but we prefer our wording and prefer to retain it if possible.
4	Results: Results can be written much more concise. There is no need to repeat all the data given in table 2.	Thank you. In general, our approach to the reporting of results, is to state all key findings in the text. This will inevitably duplicate some of the table contents, but we would argue that there are many other statistics in the table that do not appear in the text.
5	Discussion : The second limitation can be rephrased as “low response rate.... affects the external validity (generalizability) of the study results”	Low responses rates do not necessarily affect generalisability or validity. This would be the case if the sample was unrepresentative. In fact, we have a good representative sample. In the smaller units however, we only have 1/3 of units responding – this is understandable given that smaller units have less resource to give time to responding to surveys. Again we prefer to retain our wording not least because the future trial would take place in the level 1 and 2 units who start and build up feeds in the infants of infants for the trial.
6	What the study adds : Points 2 & 3 need to be rephrased	Thank you. We have re-worded this now.

Reviewer 2:

“With this survey, the authors attempted to summarize current feeding practices concerning gastric residual volumes in preterm infants. The authors conducted this research to examine the feasibility of a future trial with a study arm in which clinicians do not measure gastric residual volumes. The conclusion statement suggests that a randomized trial with such characteristics is feasible. Although this reviewer may not agree on all the issues raised in the article, I praise the authors for their efforts on describing current feeding practices in the UK. Below I summarized my specific comments and suggestions.”

Reviewer 2 Comments		Response
Major		
1	This article would have been more persuasive if the authors had related their findings with NEC outcomes of the units included in this survey. Specifically, it would have been interesting to know if units that do not currently measure GRVs have a lower incidence of NEC or culture-proven sepsis.	Thank you – this point is well made. The reason this additional information is not included here, is that the e-survey’s specific objective was to describe practice. Drawing inference linking practice with outcomes is reserved for a future clinical trial. The e-survey presented in this paper forms part of a wider piece of work (the GASTRIC study) looking at the feasibility of a trial in the UK.

2	A correlation analysis between survey responses of senior doctors and survey responses of nurses from the same unit would help validate the accuracy of their responses. In survey-based methodology, some suspect that senior doctors report what they think they do and nurses and other front-line clinicians report what they really do.	Each unit completed a single survey, but this was completed together at the same time by a team of clinicians ideally including a doctor, a nurse and a dietitian (see Methods section). Therefore we are unable to look at responses split by clinician type. While we understand that opinions about the relevance of GRV will differ between types of clinician, the main aim here was to describe what current practice is to inform the potential future trial.
3	In the discussion section, it would be helpful to make the distinction between practices that promote routine GRV measurements and practices that promote selective GRV measurements. The practice of not performing GRV measurements at all should be discussed separately.	We compared the routine use of GRV measurement with not measuring as this was the brief from the funder and what we were interested in for a future trial. The discussion is therefore written with these two GRV approaches in mind and to add detail about selecting which infants to use it for would have been confusing. There is detail on this in the supplementary material but we do not agree that it should feature prominently in the discussion.
4	One aim of the survey was to define the control arm of a future trial in which GRVs are not measured. The characteristics of this potential control arm were not reported. Based on the study findings and the limited evidence available, it would seem reasonable to have a control arm with selective GRV measurements. In this group, GRVs would be routinely returned regardless of color to maintain a low gastric pH and minimize the risk of bacterial colonization.	Please see the previous answer. This work took place before a whole Delphi consensus process was undertaken with staff and parents which looked at which group of infants should be included in the proposed trial and this work is presented elsewhere. We don't agree that the control arm should include selective measurements. The proposed future trial would compare no measurement with routine measurement as standard approaches. Routine measurement is the typical approach in UK units that use GRV to guide practice. It could also be very difficult to show a difference in terms of trial outcomes between a selective measurement approach and a non-measurement approach which would also be very difficult to operationalise in a trial. We believe this approach is therefore undesirable for the future trial and would be confusing if raised in the discussion.
5	The proposed trial implies an effort to standardize the technique of gastric content aspiration among preterm infants. A clear efficacy endpoint of checking GRVs to prevent adverse outcomes of preterm infants should be stated to make the safety endpoint relevant. As currently written, the justification for a trial with safety as the primary outcome is weak.	Our conclusion and the "What this study adds" section clearly define what we feel this particular study adds to the literature, especially the sentence that has been reworded for clarity and to now says; 'The heterogeneity of approaches regarding GRV measurement supports the need for a randomised trial to enable an evidence-based approach to the practice'. We do not propose what the trial should look like beyond the interventions as the work described elsewhere clarified what

		the trial should look like. This is mentioned at the end of the methods section and will be referenced in the final, published version of the manuscript.
--	--	--

VERSION 2 – REVIEW

REVIEWER	Reviewer name: Thangaraj Abiramalatha Institution and Country: Department of Neonatology Sri Ramachandra Institute of Higher Education and Research, Chennai, India. Competing interests: None
REVIEW RETURNED	22-May-2020

GENERAL COMMENTS	1. Though the study is a survey of practice in NICUs and the population of neonates is not described, gastric residual monitoring is done usually only in preterm infants and term infants undergone GI surgery. The authors can at the minimum report the answer for the question "Are gastric residual volume measured for all babies, or below a set gestation/birth weight or a specific condition", which is there in their questionnaire.
---

REVIEWER	Reviewer name: Ariel Salas Institution and Country: University of Alabama at Birmingham, Birmingham, AL, USA Competing interests: None to disclose relevant to this article
REVIEW RETURNED	24-May-2020

GENERAL COMMENTS	No additional comments
------------------------

VERSION 2 – AUTHOR RESPONSE

Reviewer: 1

Comments to the Author

1. Though the study is a survey of practice in NICUs and the population of neonates is not described, gastric residual monitoring is done usually only in preterm infants and term infants undergone GI surgery. The authors can at the minimum report the answer for the question "Are gastric residual volume measured for all babies, or below a set gestation/birth weight or a specific condition", which is there in their questionnaire.

Response;

We did ask the question. "Are Gastric Residual Volume measured for all babies, or just below a set gestational age/birth weight or for a specific condition"?

Unfortunately being an open ended question this question elicited a lot of mixed responses that are hard to easily summarise. We have therefore now provided some detail on this in the paper (in the results and in the discussion) and submitted added a new supplementary table with the responses in table 1. We have also described this as a weakness of our study in the discussion.

Result section sentences

90 open text responses were received to the question "Are Gastric Residual Volume measured for all babies, or just below a set gestational age/birth weight or for a specific condition"? Over 30 responses said all babies should have gastric aspirates measured, with some additional responses saying all but limiting this to those on gastric tube feeds or until full feeds are established. Just six responses mentioned a gestational age cut off, four suggesting <32 weeks, one <27 weeks and one <34 weeks gestation. Just one responses indicated a birthweight criterion (under 1500g at birth). Supplementary table 1 presents the responses.

Discussion sentences

Thirdly, we asked an open rather than a closed question to seek detail on which babies (in terms of gestation, birthweight or conditions) have residual volumes measured which made the data hard to analyse. Further details were obtained in related research and have been published elsewhere (18).

We have also added reference and acknowledgement to our now published NIHR HTA journal paper (ref 18).

We hope this is all acceptable and look forward to hearing from you.

Lastly, Louise Roper's work details have been amended as she has changed Departments in the University of Liverpool